# Thermochemical and Enzymatic Saccharification of Water Hyacinth Biomass into Fermentable Sugars

Evelyn Romero-Borbón [1], Andrea Edith Oropeza-González [1], Yolanda González-García [2] and Jesús Córdova [1,*]

1   Departamento de Química, Centro Universitario de Ciencias Exactas e Ingenierías, Universidad de Guadalajara, Blvd. Gral. Marcelino García Barragán 1421, Col. Olímpica, Guadalajara C.P. 44430, Jalisco, Mexico; evelynrom@gmail.com (E.R.-B.); andiieorogon@gmail.com (A.E.O.-G.)
2   Departamento de Madera, Celulosa y Papel, Centro Universitario de Ciencias Exactas e Ingenierías, Universidad de Guadalajara, Carretera Guadalajara-Nogales km 15.5, Zapopan C.P. 45220, Jalisco, Mexico; yolanda.ggarcia@academicos.udg.mx
*   Correspondence: antonio.cordova@academicos.udg.mx

**Abstract:** Water hyacinth (WH) is a free-floating perennial aquatic plant that is considered a pest, due to its rapid grown rate and detrimental effects on environment and human health. It is nearly impossible to control WH growth, with mechanical extraction being the most acceptable control method; nevertheless, it is costly and labor-intensive. WH lignocellulosic biomass represents a desirable feedstock for the sustainable production of liquid fuels and chemical products. In this work, optimal conditions of thermochemical pretreatment for the release of reducing sugars (RS) from WH biomass were established: 0.15 mm of particle size, 50 g of dried solid/L of $H_2SO_4$ (3% *w/v*) and 20 min of heating time at 121 °C. Applying this pretreatment, a conversion of 84.12% of the hemicellulose fraction in the raw WH biomass into reducing sugars ($277 \pm 1.40$ mg RS/g DWH) was reached. The resulting pretreated biomass of WH (PBWH) was enzymatically hydrolyzed by using six enzymatic complexes (all from Novozymes). Among them, NS22118 (beta-glucosidase) and Cellic® CTec2 (cellulase and hemicellulose complex) achieved higher saccharifications. By using NS22118 or a mixture of NS22118 and Cellic® CTec2, PBWH conversion into RS was complete. Monosaccharides released after pretreatment and enzymatic hydrolysis were mostly pentoses (arabinose and xylose) and hexoses (glucose), respectively.

**Keywords:** water hyacinth; enzyme saccharification; thermochemical pretreatment

## 1. Introduction

Water hyacinth (*Eichhornia Crassipes*) is a native plant from the Amazon basin that has spread to many other tropical and sub-tropical regions, invading bodies of water [1,2]. Water hyacinth (WH) is the world's most aggressive free-floating perennial aquatic plant (Hydrophyte) and can cover an entire aquatic body in a thick, compact carpet in a two-to-three week period [2–4]. This thick carpet blocks the sunlight from reaching below the water's surface, interfering with the growth of other aquatic organisms and eliminating native species [4–6]. WH is considered a pest in the aquatic environments, acting as a breeding ground for disease vectors [7].

Managing WH through chemical, biological and mechanical controls has not proven effective and causes various complications [8]. Herbicides used as chemical control (e.g., glyphosate, 2,4-D) pollute water bodies and eliminate non-target organisms [4]. Introducing predators (e.g., weevil beetles) as agents of biological control has potentially catastrophic consequences for the environment [8]. While mechanical removal remains the most useful to control the WH population, it is costly and labor-intensive [9]. Furthermore, once WH biomass is harvested, if it is not properly managed, it can cause ecological problems in the locations where discarded. In this context, an outstanding number of studies have investigated the conversion and potential use of WH biomass into value-added products,

suggesting a valorization of WH [10–14]. The problem of solid waste management can potentially be transformed into an opportunity to make mechanical removal economically feasible and profitable [4].

Another important use of water hyacinth is phytoremediation. Indeed, WH is used for the removal of pollutants due to its ability to grow in highly polluted waters [15–20]. Moreover, WH can tolerate substantial variations in the types and concentrations of nutrients and pollutants, pH levels (optimum growth at pH 6–8) and temperatures (from 1 to 40 °C, optimum growth at 25–30 °C). The characteristics cause WH to be considered a pest (rapid growth rate with extensive root system, high biomass yield, high tolerance to pollutant toxicity and physicochemical conditions, no food potential for animals or humans, the lack of effective control in plant populations), could help WH's bioremediatory role in polluted water bodies.

Water hyacinth is composed (on dry weight) of cellulose (18.2 to 19%), hemicellulose (48.7 to 50%), lignin (3.5 to 3.8%) and crude protein (13 to 13.5%) [21]. However, it is worth noting that this composition may vary depending on the geographical area and climatic conditions [22]. Due to its composition, WH lignocellulosic biomass is considered a desirable feedstock for the sustainable production of liquid fuels and chemical products through the biorefinery processes [10–14]. Additionally, as an aquatic plant, it does not compete with agricultural crops for land use.

In lignocellulosic materials, cellulose forms highly crystalline microfibrils embedded in a hemicellulose and lignin matrix. Due to the recalcitrant nature of lignocellulosic biomass, a pre-treatment is needed in order to open up the fibers and decrease the crystallinity of cellulose [23], thereby increasing its accessibility to saccharification enzymes [24–26]. Based on the particular composition of vegetal biomass, pretreatment methods need to be adapted for each biomass type [23]. Vegetal biomass pretreatments can be classified into four categories: (i) physical methods involve fragmentation processes (milling, grinding, drying), sonication and pyrolysis; (ii) chemical methods include acid and alkali treatments, oxidation, ionic liquids and organosolv pretreatment; (iii) biological methods include bacterial, fungi and enzyme treatments; (iv) physicochemical methods comprise steam explosion, ammonia fiber explosion, among others [27–29]. Acid-thermal pretreatment is one of the most widely used, due to simplicity and efficient performance, removing the lignin portion, hydrolyzing hemicellulose at relatively moderate temperatures, with high sugar yields and low formation of degradation compounds, and enhancing the accession of carbohydrolase enzymes to inner space of the pretreated biomass [30–35]. Regarding the thermochemical pretreatment of WH biomass, the reported hydrolysis conditions (solid loading, hydrolysis temperature, heating time and particle size) and consequently, the obtained reducing sugar yields are variable [33–36]. Those reported ranges of hydrolysis conditions were taken into consideration, being optimized in this work.

After pretreatment, biomass can be enzymatically hydrolyzed. However, cellulose hydrolysis remains a main limiting factor for the efficient utilization of lignocellulose [37,38]. Multiple enzymatic activities are needed to hydrolyze cellulose into soluble sugar monomers [39,40]. At least three major types of enzymes are required for hydrolyzing cellulose: endoglucanase (EG), exoglucanase (cellobiohydrolase CBH) and β-glucosidase (BGL) [24]. To date, studies in enzymatic hydrolysis of pretreated HW biomass have not yielded complete conversion to monosaccharides [41–44].

The aim of this work was to study optimal conditions for the complete saccharification of water hyacinth biomass by the process of physicochemical pretreatment, followed by an enzymatic treatment, with the objective of encouraging its sustainable utilization.

## 2. Materials and Methods

### 2.1. Sampling and Treatment of Water Hyacinth

Water hyacinth plants were collected in February and August 2020 from the shores of Lake Chapala in the Mexican State of Jalisco (20°17′24.5″ N 103°11′44.3″ W). Plants were washed with tap water to eliminate dirt and then fractioned in three parts: root, leaves and

stems. The wet plants were weighed, registered for each fraction, cut into smaller pieces, and dried at 70 °C in a hot air oven (Terlab, Mexico) for 72 h until constant weight was reached. Dry weights were registered and moisture contents were calculated. Dried stems were milled in a blade mill (Vayco, Mexico) and sieved through 8, 20 and 100 Tyler meshes, obtaining particle sizes of 2.36, 0.85 and 0.15 mm, respectively.

### 2.2. Analysis of Water Hyacinth Biomass Composition

Water hyacinth biomass composition was analyzed, according to Browning [45], Technical Association of the Pulp and Paper Industry (TAPPI) and American Society for Testing and Materials (ASTM) norms: ash content, organic solvent extraction materials, water extractable materials and acid insoluble lignin. Holocellulose was quantified according to the test method for holocellulose in wood (ASTM D-110-56) [46]. Hemicellulose was determined by weight difference between cellulose and holocellulose.

### 2.2.1. Material Extractable in Organic Solvents

Test method for solvent extractives of wood and pulp (T-204 om-88) was used by means of a Soxhlet system and hexane as solvent. For the analysis, 10 g of each sample were used [47].

### 2.2.2. Material Extractable in Water

Test method for water solubility of wood and pulp (T-207 om-93) was used in the residue resulting from extraction with solvents [48].

### 2.2.3. Ash Content

It was determined in accordance with ash in wood, pulp, paper and paperboard, test method (T-211 cm-93), for which 1 g of sample was used [49].

### 2.2.4. Holocellulose Content

One hundred milliliters of 1.5% (*w/v*) sodium chlorite dissolved in glacial acetic acid were added to 2 g of sample. The reaction was carried out at 75 °C for 5 h. The mixture was then filtered in a Gooch crucible, washed with acetone and dried at 105 °C for 1 h. The percentage of holocellulose was determined by the difference between the initial and final weight. All samples were analyzed in triplicate and the results were expressed as a percentage on a dry basis [22].

### 2.3. Analysis of Sugars

Reducing sugars (RS) were assayed using the dinitrosalicylic acid (DNS) reagent [50]. Tubes containing 100 µL of sample and 100 µL of DNS reagent were mixed and placed in a boiling water bath for five minutes. Reaction was stopped by placing tubes in an ice-water bath and 800 µL of distilled water were added to each tube. Absorbance was measured at 540 nm in a UV-Vis spectrophotometer (GENESYS 10, Thermo Electron Corporation, Madison, WI, USA). Standard curves of glucose and xylose were prepared.

Monosaccharides from water hyacinth (WH) hydrolysates were identified and quantified by high performance liquid chromatography (HPLC, Waters®, Milford, MA, USA) (Waters 600 system) with refractive index detector model 2414. An Aminex HPX-87P column (BioRad, Dubai, United Arab Emirates) was operated at 80 °C and 0.6 mL/min flux, with HPLC water as the mobile phase. Injection volume was 20 µL. Samples were diluted 1:10 and filtered by 0.22 µm membranes before being injected. Glucose, xylose, arabinose, galactose and mannose standards were prepared at 1 g/L.

### 2.4. Thermochemical Pretreatment of Water Hyacinth

Dry water hyacinth (DWH) was pretreated using 3% *w/v* $H_2SO_4$ at 121 °C. The thermochemical pretreatment was optimized by studying the effect of solid load (10, 20 and 50 mg of DWH/mL $H_2SO_4$), reaction times (10, 20, 40 and 60 min) and particle sizes (0.15, 0.85 and

2.36 mm) on the release of reducing sugars (RS). Tubes containing WH biomass suspended in 3% *w/v* $H_2SO_4$, were vigorously mixed in vortex for 20 s. Heating was carried out in an autoclave (MELAG Type 23, MELAG Medizintechnik GmbH & Co. KG, Berlin, Germany) at 121 °C at different reaction times, followed by a sudden decompression. Once cooled, pH was neutralized by adding 2N NaOH and mixing in vortex. Suspensions were centrifuged at 4000 rpm ($2235 \times g$) for 15 min in a Firlabo centrifuge (Nantes, France). Reactions were carried out in duplicate. Reducing sugars on supernatants were assayed in triplicate.

The pellet was washed with distilled water and vacuum filtered through cellulose filter paper (Whatman No. 40). This solid was dried to constant weight (at 80 °C for 24 h), milled with a mortar and sieved through a 200 mesh. The obtained powder was the pretreated biomass of WH (PBWH).

### 2.5. Enzymatic Assays

Enzymatic activities from the six commercial cocktails were measured, employing carboxymethylcellulose (CMC) and birchwood xylan (BX) as substrates. Enzyme activities were determined by mixing 20 µL of a properly diluted enzyme sample with 180 µL of a substrate solution (1% CMC or BX in 0.1 M citrate buffer, pH 4.8) at 50 °C for 30 min. Reactions were stopped, adding 40 µL 2 N NaOH and mixing in vortex. RS concentration was assayed by using the dinitrosalicylic acid (DNS) reagent. Glucose and xylose standard curves were prepared to determinate cellulases and xylanases activities, respectively. One unit of enzyme activity was defined as the amount of enzyme required to release 1 µmol of RS per minute.

### 2.6. Enzymatic Hydrolysis of Pretreated Biomass of Water Hyacinth

Pretreated biomass of WH (PBWH) was saccharified, using six commercial carbohydrolase complexes (from Novozymes): cellulase NS22086, endo-xylanase NS22083, beta-glucosidase NS22118, arabinose, beta-glucanase, cellulase, hemicellulase, pectinase and xylanase complex NS22119, beta-glucanase and xylanase complex NS22002 and cellulase and hemicellulose complex Cellic® CTec2 (Table 1). Initially, enzyme complexes were diluted 1:100 in citrate buffer (0.1 M pH 4.8), containing sodium azide (20 mg/L). Two solid loads (50 and 125 mg PBWH/mL diluted enzyme solution) were assayed. Tubes were incubated at 50 °C and 140 rpm for 36 h. Reactions were stopped by placing tubes in an ice-water bath, adding 200 µL of 2 N NaOH and mixing in a vortex. Suspensions were centrifuged at 4000 rpm ($2235 \times g$) for 15 min. For comparative purposes, this experiment was repeated using DWH (not pretreated biomass).

**Table 1.** Commercial carbohydrolase complexes employed in this work.

| Commercial Name | Enzymatic Activities | pH | Density (g/mL) | Temperature (°C) | Activity [1] |
|---|---|---|---|---|---|
| NS22086 | Cellulase | 5.0–5.5 | 1.15 | 45–50 | 1000 BHU/g |
| NS22083 | Endo-xylanase | 4.5–6.0 | 1.09 | 35–55 | 2500 FXU-S/g |
| NS22118 | Beta-glucosidase | 2.5–6.5 | 1.2 | 45–70 | 250 CBU/g |
| NS22119 | Arabinase, beta-glucanase, cellulase, hemicellulase, pectinase and xylanase complex | 4.5–6.0 | 1.19 | 25–55 | 100 FBG/g |
| NS22002 | Beta-glucanase and xylanase complex | 5.0–6.5 | 1.2 | 40–60 | 45 FBG/g |
| Cellic® CTec 2 | Cellulases and hemicellulases complex | 5.0–5.5 | 1.3 | 45–55 | 1000 U/g |

[1] BHU: biomass hydrolysis unit, CBU: cellobiase unit, FBG: fungal beta-glucanase unit, FXU: xylanase unit.

Once the best solid load was established, several dilutions (1:10, 1:50, 1:100 and 1:1000) of Cellic® Ctec2 (Cellulase and hemicellulose complex) were performed, in order to determine the minimum concentration of carbohydrolase complex to obtain the highest RS conversion yield from PBWH, under the following reaction conditions: 5% solid load, pH 4.8, 50 °C and 40 rpm for 24 h. For this Cellic® CTec2 minimum concentration, enzyme activity was assayed against CMC as substrate and activities of all enzyme commercial

preparations were diluted and standardized to reach 5.62 U/mL. With the standardized enzymes activities, PBWH were saccharified for 36 h.

In order to minimize commercial enzyme concentrations, several dilutions of beta-glucosidase NS22118 (1:10, 1:50 and 1:100) were performed to hydrolyze PBWH and two enzyme complexes (NS22118 and Cellic® CTec2) were mixed to find synergistic effects in the saccharification of PBWH. Blanks were prepared with water instead of enzyme solution. Reactions were carried out in duplicate and RS were assayed in triplicate. Data represent mean and standard deviation (*n* = 6). Monosaccharides were measured by HPLC.

### 2.7. Statistical Analysis

Results were analyzed by the analysis of variance (ANOVA) tested at $\alpha \leq 0.05$ significance level, using Statgraphics Centurion XV® software.

### 2.8. Yield Estimates

Maximum theoretical yield was calculated considering the ratio of cellulose/DWH (% *w/w*) or hemicellulose/DWH (% *w/w*) in WH stems. The glucose and xylose weight ratio with respect to the molecular weight of cellulose and hemicellulose is 1.111 and 1.136, respectively.

$$Maximum\ theoretical\ yield = 1.111 \frac{g\ of\ glucose}{g\ of\ cellulose} \times \frac{19.55\ g\ of\ cellulose}{100\ g\ of\ DWH} + 1.136 \frac{g\ of\ xylose}{g\ of\ hemicellulose} \times \frac{28.99\ g\ of\ hemicellulose}{100\ g\ of\ DWH}$$

where DWH stands for dry water hyacinth. The equation to determine the saccharification yield was as follows:

$$Saccharification\ yield\ (\%) = \frac{Experimental\ yield}{Maximum\ theoretical\ yield} \times 100$$

## 3. Results and Discussion

### 3.1. Water Hyacinth Biomass Composition

Plant fractions dry matter percentages are reported in Table 2. Cellulose and hemicellulose contents were higher in leaves and stems, respectively (Table 3). On the other hand, lignin content was lower in stems (3.3%), making them a suitable choice for saccharification. Furthermore, because stems represented most of the WH biomass (53.2% of the total weight, Table 2), they were selected for chemical pretreatment and enzymatic hydrolysis experiments. It is worthy to mention that the polymers content in the overall WH biomass (first and second harvest) (Table 3) is within the values reported for WH collected in India, China and Kenya: cellulose, 18–33%; hemicellulose, 23–48.7%; and lignin, 1.1–9%, respectively [21,51,52].

**Table 2.** Moisture and dry matter content of water hyacinth fractions.

| Harvest | Plant Section | Fresh Plant (%) | Dry Plant (%) | Moisture (%) | Dry Weight (%) |
|---|---|---|---|---|---|
| **First** | Leaf | 10.12 ± 4.0 | 17.14 ± 3.3 | 82.96 ± 5.12 | 17.04 ± 1.05 |
| | Stem | 55.53 ± 6.5 | 44.96 ± 6.8 | 95.15 ± 0.15 | 4.85 ± 0.01 |
| | Root | 34.35 ± 5.9 | 37.9 ± 5.6 | 92.71 ± 0.68 | 7.29 ± 0.05 |
| **Second** | Leaf | 23.89 ± 0.17 | 37.22 ± 0.12 | 87.46 ± 1.58 | 12.54 ± 0.23 |
| | Stem | 50.86 ± 6.02 | 44.62 ± 5.72 | 93.63 ± 0.51 | 6.37 ± 0.03 |
| | Root | 25.24 ± 0.04 | 18.17 ± 0.42 | 92.34 ± 1.37 | 7.66 ± 0.11 |

First harvest was in February and Second harvest in August 2020. Data represent the mean and the standard deviation of three assays.

**Table 3.** Characterization of water hyacinth composition (% dry weight) from Lake Chapala.

| Harvests | Fraction | Cellulose | Hemicellulose | Ash | HE | HWS | ASL |
|---|---|---|---|---|---|---|---|
| | Leaf | 32.84 ± 0.50 | 24.17 ± 1.45 | 10.32 ± 0.23 | 20.53 ± 0.23 | 20.59 ± 0.20 | 6.22 ± 0.03 |
| **First** | Stem | 20.50 ± 0.10 | 30.60 ± 0.02 | 13.36 ± 0.12 | 21.47 ± 0.11 | 15.46 ± 0.09 | 3.21 ± 0.08 |
| | Root | ND | ND | 15.30 ± 0.09 | 8.37 ± 0.08 | 7.63 ± 0.02 | 8.31 ± 0.07 |
| | Leaf | 30.25 ± 6.03 | 22.37 ± 2.30 | 11.15 ± 1.13 | 20.27 ± 0.02 | 21.39 ± 0.08 | 6.57 ± 0.24 |
| **Second** | Stem | 18.60 ± 0.35 | 27.38 ± 2.12 | 13.11 ± 0.23 | 21.71 ± 0.16 | 15.87 ± 0.15 | 3.33 ± 0.06 |
| | Root | ND | ND | 12.52 ± 5.49 | 8.47 ± 0.10 | 7.63 ± 0.02 | 8.55 ± 0.14 |

ND: not determined; HE: hexane extractable; HWS: hot water soluble; ASL: acid soluble lignin. First harvest was in February and Second harvest in August 2020. Data represent the mean and the standard deviation of three assays.

*3.2. Thermo-Chemical Pretreatments of Water Hyacinth Biomass*

A thermo-chemical pretreatment of 121 °C and $H_2SO_4$ (3% *w/v*) was chosen to break down the structural complexity of lignocellulosic biomass of dry water hyacinth (DWH). This pretreatment was optimized by modifying the following variables of experimental conditions: particle size (0.15, 0.85 and 2.36 mm), reaction time at 121 °C (10, 20, 30 and 40 min) and dry solid loading (10, 20 and 50 g DWH/L $H_2SO_4$ at 3% *w/v*). The variable response of these chemical pretreatments was the reducing sugars (RS) released from DWH.

The proposed chemical pretreatment ($H_2SO_4$ at 3% *w/v* and 121 °C) proved to be efficient to break down the structural complexity of lignocellulosic biomass of water hyacinth as it has been reported by others works [53–56]. All tested variables were significant ($\alpha \leq 0.05$) to describe reducing sugars concentration released from DWH. Optimal conditions of chemical pretreatment to obtain maximal concentrations of RS (9.9 and 9.95 g/L) were found at the smallest particle size (0.15 mm), the highest solid loading (50 g/L) and reaction times of 20 and 40 min, respectively (Figure 1). It is worth noting that before assaying RS in the hydrolysates, pH was neutralized by adding 2 N NaOH and consequently, reported values in Figure 1 consider the volume added of 2 N NaOH. Furthermore, no significant differences ($\alpha \leq 0.05$) were found for RS maximal concentrations for 20 or 40 min of reaction. Consequently, 20 min was selected as the best reaction time from an energy saving point of view. In addition, less furfural could be generated during this heating time.

Since water hyacinth stems are mostly composed by hemicellulose (Table 3), chemical pretreatment yielded monosaccharides mostly constituted by pentoses (arabinose and xylose) and followed by hexoses (glucose and galactose) (Figure 2). It is well known that hemicellulose hydrolyses faster than cellulose in an acid environment, due to its chemical characteristics [57]. By applying this chemical pretreatment to water hyacinth biomass, only 40% (*w/w*) of the solid matter was recovered. The resulting pretreated biomass was mostly constituted of cellulose.

It is important to highlight that in the thermo-chemical pretreatment of water hyacinth biomass, particle size, solid load and reaction time played a significant role in opening up the fibers, decreasing the cellulose crystallinity and breakdown of the hemicellulose portion in WH biomass, as previously reported [35].

In addition, at reaction times of 10, 20 and 40 min (at 121 °C), a fraction of other carbohydrates different from monosaccharides were revealed by chromatographic analysis (certainly oligosaccharides). This carbohydrate fraction disappeared at the highest reaction time assayed (60 min at 121 °C) and it was converted equivalently into more monosaccharides (Figure 2).

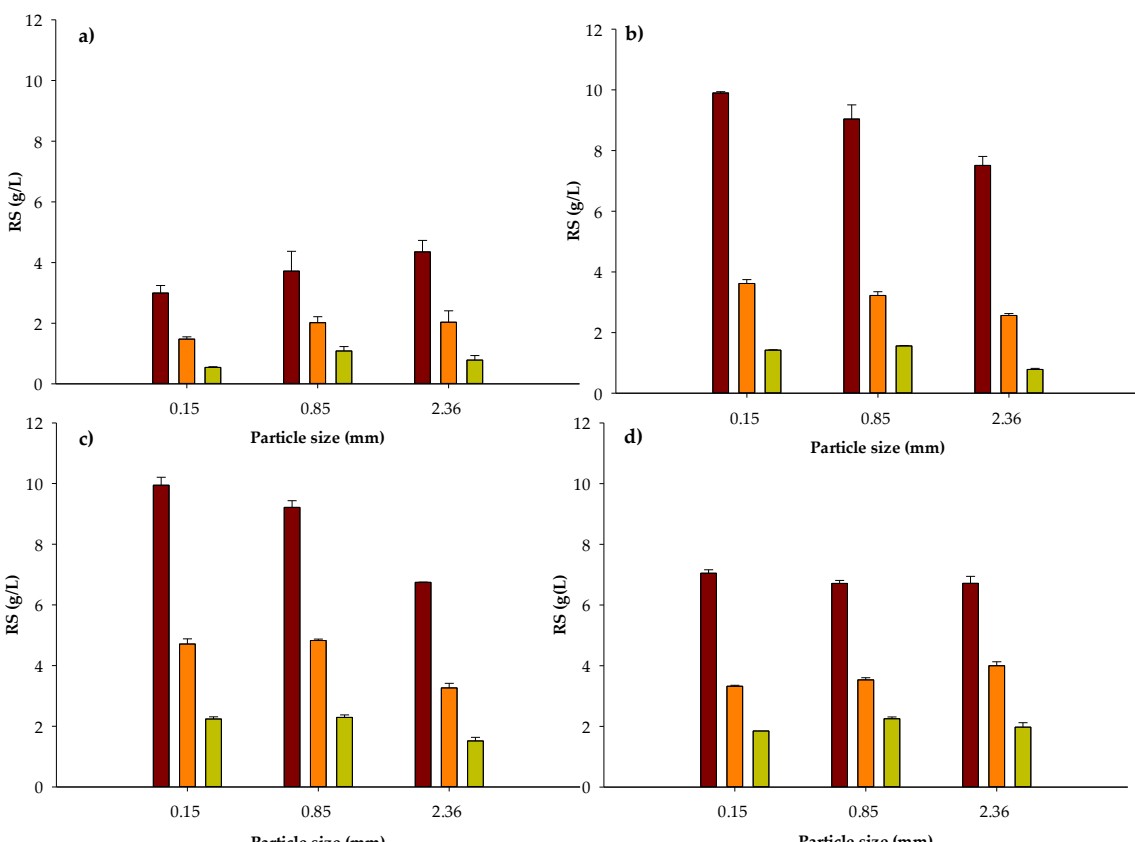

**Figure 1.** Released RS after thermo-chemical pretreatment of water hyacinth biomass at different particle sizes (0.15, 0.85 and 2.36 mm), reaction times ((**a**) 10 min, (**b**) 20 min, (**c**) 40 min and (**d**) 60 min) and solid loading (50 ■, 20 ■ and 10 ■ g water hyacinth biomass/L $H_2SO_4$ 3% *w/w*). Data represent the mean and the standard deviation of three assays.

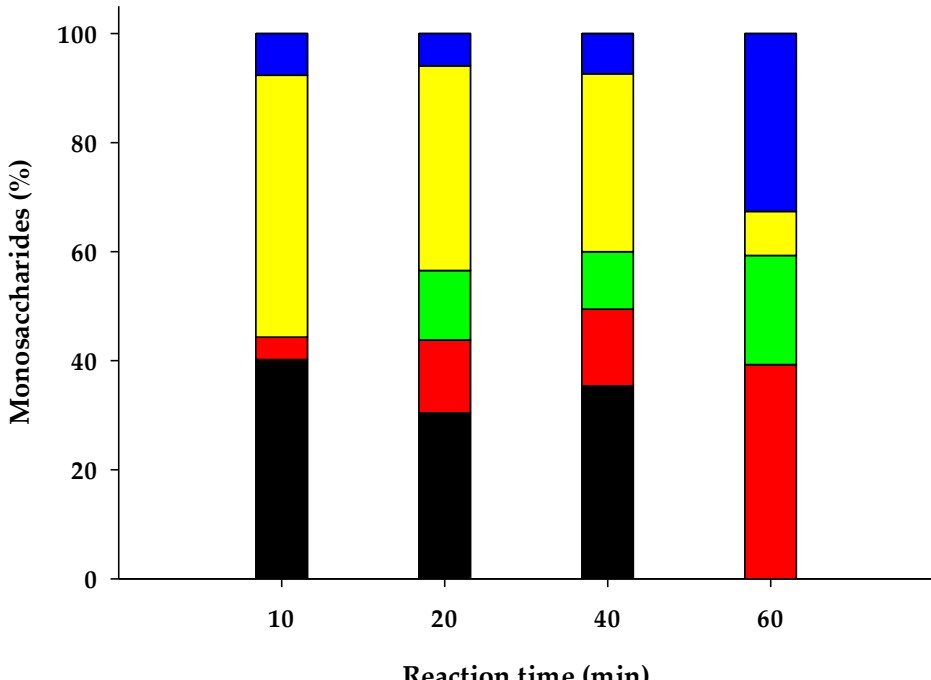

**Figure 2.** Composition of monosaccharides on thermo-chemical pretreated water hyacinth hydrolysates by HPLC analysis. Glucose ■, Xylose ■, Galactose ■, Arabinose ■ and Unknown ■.

### 3.3. Thermochemical Pretreatment Efficiency

A yield of 277.04 $\pm$ 1.40 mg RS/g DWH was calculated for optimal conditions of the chemical pretreatment (0.15 mm of particle size, 50 g DWH/L of $H_2SO_4$ 3% of solid loading and 20 min of reaction time). This yield value represented a conversion efficiency of 84.12% when compared with the theoretical yield (329.33 mg RS/g DWH) for hemicellulose portion in DWH. The experimental yield value obtained in this research was similar to others values reported by other workers for thermo-chemical pretreatments of DWH, using sulfuric acid [35,36,58].

### 3.4. Enzymatic Saccharification of Chemically Pretreated Water Hyacinth Biomass

Six commercial preparations of carbohydrolase from Novozymes (NS22083, NS22086, NS22118, NS22119, NS22002 and Cellic® CTec2) were evaluated for their hydrolytic capacity of the pretreated biomass of water hyacinth (PBWH).

Initially, a dilution 1:100 for each commercial enzyme was assayed for two solid loadings (50 and 125 g PBWH/L of diluted commercial enzymes). A solid loading of 50 g PBWH/L was better to reach maximal RS concentrations. The highest RS yields (174.14 $\pm$ 28.81 and 162.66 $\pm$ 54.03 mg RS/g PBWH) were achieved using endo-xylanase NS22083 and cellulase Cellic® CTec2, respectively (Table 4).

**Table 4.** Enzymatic hydrolysis of pretreated and not pretreated biomass of water hyacinth (at two solid loadings) by diluted (1:100) commercial enzymes.

| Solid Loading | Commercial Enzyme | Pretreated Biomass | | Not Pretreated Biomass | |
|---|---|---|---|---|---|
| | | RS Concentration (g/L) | RS Yield (mg/g PBWH) | RS Concentration (g/L) | RS Yield (mg/g DWH) |
| 50 | NS22086 | 4.52 $\pm$ 0.00 | 108.11 $\pm$ 1.68 | 0.95 $\pm$ 0.26 | 22.85 $\pm$ 6.48 |
| | NS22083 | 7.29 $\pm$ 1.22 | 174.14 $\pm$ 28.8 | 0.58 $\pm$ 0.05 | 14.00 $\pm$ 1.32 |
| | NS22118 | 2.53 $\pm$ 0.00 | 60.51 $\pm$ 0.12 | 0.48 $\pm$ 0.15 | 11.46 $\pm$ 3.53 |
| | NS22119 | 4.65 $\pm$ 0.08 | 111.73 $\pm$ 2.14 | 1.14 $\pm$ 0.05 | 27.53 $\pm$ 1.11 |
| | NS22002 | 2.05 $\pm$ 0.08 | 49.21 $\pm$ 2.03 | 0.25 $\pm$ 0.02 | 6.01 $\pm$ 0.47 |
| | Cellic® CTec2 | 6.82 $\pm$ 2.31 | 162.66 $\pm$ 54.0 | 1.16 $\pm$ 0.17 | 27.78 $\pm$ 4.01 |
| 125 | NS22086 | 5.16 $\pm$ 0.00 | 49.46 $\pm$ 0.15 | ND | ND |
| | NS22083 | 2.99 $\pm$ 0.33 | 28.78 $\pm$ 3.12 | ND | ND |
| | NS22118 | 0.57 $\pm$ 0.08 | 5.49 $\pm$ 0.81 | ND | ND |
| | NS22119 | 0.47 $\pm$ 0.01 | 4.55 $\pm$ 0.10 | ND | ND |
| | NS22002 | 1.70 $\pm$ 0.80 | 16.30 $\pm$ 7.64 | ND | ND |
| | Cellic® CTec2 | 3.50 $\pm$ 0.60 | 33.70 $\pm$ 5.79 | ND | ND |

Reaction conditions: 50 °C and 140 rpm for 24 h. DWH is dry water hyacinth (not pretreated biomass). PBWH is pretreated biomass of water hyacinth. Solid loadings are g PBHW/L of 1:100 diluted commercial enzyme. RS: reducing sugars. ND: not determined. Data represent the mean and the standard deviation of three assays.

For comparative purposes, the WH biomass without chemical pretreatment was used to be hydrolyzed, using the diluted (1:100) enzymatic extracts. For not pretreated water hyacinth biomass, all of the diluted commercial enzyme preparations showed lower activities, revealing the importance of chemical pretreatment of the water hyacinth biomass. However, as observed in Table 4, NS22086, NS22119 and Cellic® CTec2 had notable hydrolytic activities of the not pretreated biomass, representing the 13.1%, 15.8% and 15.9%, respectively, compared to the best value obtained (174.14 mg AR/g DWH), using the pretreated biomass and NS22083.

Afterwards, several dilutions (1:10, 1:50, 1:100 and 1:1000) of the Cellic® CTec2 (cellulase and hemicellulase complex) were performed, in order to determine the best concentration to obtain the highest RS conversion yield from chemically pretreated biomass of water hyacinth (PBWH), under same reaction conditions (50 g PBHW/L of diluted enzyme complex, pH 4.8, 50 °C, 140 rpm) for 24 h. Cellic® Ctec2 1:50 dilution was selected due to its high hydrolysis value (414.40 $\pm$ 1.13, mg RS/g PBWH, Table 5). For Cellic® Ctec2 1:50 dilution, enzyme activity was assayed against CMC, obtaining 5.62 U/mL. From this knowledge, all enzyme commercial preparations were diluted to reach 5.62 U/mL,

using CMC as the substrate. By using these standardized enzyme activities and reaction conditions, pretreated biomass of water hyacinth was saccharified for 36 h (Table 6).

**Table 5.** Saccharification of pretreated biomass of water hyacinth (PBWH) by using Cellic® CTec2, NS22118 and mixtures of Cellic® Ctec2 and NS22118 at several dilutions.

| Enzyme | Dilution | RS (g/L) | RS (mg/g PBWH) |
|---|---|---|---|
| Cellic® CTec2 | 1:10 | 21.48 ± 0.78 | 514.39 ± 15.88 |
| | 1:50 | 17.42 ± 0.17 | 414.40 ± 1.13 |
| | 1:100 | 12.20 ± 0.06 | 292.31 ± 3.80 |
| | 1:1000 | 5.88 ± 0.03 | 142.58 ± 0.62 |
| NS22118 | 1:10 | 38.47 ± 2.20 | 909.51 ± 58.60 |
| | 1:50 | 2.06 ± 0.06 | 49.02 ± 0.24 |
| | 1:100 | 1.10 ± 0.04 | 26.12 ± 1.43 |
| Cellic® CTec2 (1:50) + NS22118 | 1:10 | 49.35 ± 4.04 | 1175.67 ± 84.57 |
| | 1:50 | 26.00 ± 1.23 | 616.43 ± 29.09 |
| | 1:100 | 23.21 ± 0.70 | 549.22 ± 19.67 |

Reaction conditions: 50 g PBWH/L of diluted commercial enzyme, 50 °C and 140 rpm for 36 h. In the case of Cellic® CTec2 + NS22118 mixtures, Cellic® Ctec2 dilution was fixed at 1:50, whereas NS22118 was mixed at three dilutions (1:10, 1:50 and 1:100). These diluted enzymes were mixed in a 1:1 ratio. Data represent the mean and the standard deviation of three assays.

**Table 6.** Hydrolysis of pretreated biomass of water hyacinth (PBWH) with six commercial enzyme complexes diluted enough to reach 5.62 U/mL, using carboxymethyl cellulose (CMC) as substrate.

| Enzyme Complex | Activity (U/mL) | Fold Dilution | RS (g/L) | RS Yield (mg RS/g PBWH) |
|---|---|---|---|---|
| NS22086 | 262.32 ± 0.03 | 46.66 | 17.97 ± 1.55 | 428.65 ± 33.11 |
| NS22083 | 210.50 ± 0.02 | 37.44 | 22.84 ± 0.68 | 553.57 ± 15.61 |
| NS22118 | 59.15 ± 0.05 | 10.52 | 39.60 ± 3.14 | 954.83 ± 83.91 |
| NS22119 | 116.77 ± 0.03 | 20.77 | 22.35 ± 0.40 | 534.85 ± 7.28 |
| NS22002 | 65.55 ± 0.01 | 11.66 | 8.82 ± 0.72 | 210.50 ± 15.57 |
| Cellic® CTec2 | 281.11 ± 0.05 | 50.00 | 21.66 ± 0.53 | 510.42 ± 13.97 |

Reaction conditions: 50 g PBWH/L of diluted commercial enzyme, 50 °C and 140 rpm for 36 h. Enzyme activity was assayed against CMC. Data represent the mean and standard deviation of three assays.

As shown in Table 6, NS22118 hydrolyzed more efficiently the pretreated biomass of water hyacinth (PBWH) than the other commercial enzyme preparations. In fact, NS22118 hydrolyzed almost completely PBWH (954.83 ± 83 mg of RS/g PBWH). Nevertheless, it is important to emphasize that NS22118 showed low cellulase activity (59.15 U/mL using CMC as substrate), since this enzyme complex is enriched with beta-glucosidase activity. For this reason, NS22118 was applied to hydrolyze PBWH at the highest tested concentration (1:10.52 dilution).

Lower concentrations of NS22118 (1:50 and 1:100 dilutions) were assayed to hydrolyze pretreated biomass of water hyacinth; however, yields of reducing sugars decreased as the dilution factor increased (Table 5).

In a second approach of enzymatic hydrolysis, two commercially prepared enzymes were mixed in order to increase the efficiency of the saccharification of pretreated biomass of water hyacinth (PBHW) and decrease the concentration of NS22118 used for this purpose. Cellic® Ctec2 is an enzyme complex, constituted mainly of cellulases (Table 1). This complex, whose dilution was set at 1:50, was supplemented, in a 1:1 ratio, with NS22118 at three dilutions (1:10, 1:50 and 1:100). Cellic® Ctec2 (at 1:50 dilution) and NS22118 (at 1:10 dilution) mix assured the complete hydrolysis of PBWH with a yield of 1175.67 ± 84.57 mg RS/g PBWH (Table 5). This value was very close to the theoretical yield, considering that the pretreated biomass of water hyacinth is mainly constituted of cellulose, as revealed by chromatographic analysis of the released monosaccharides. In effect, only glucose was

detected in the liquid resulting from the enzymatic reaction carried out by Cellic® Ctec2 (at 1:50 dilution) and NS22118 (at 1:10 dilution).

## 4. Conclusions

Complete saccharification of dry biomass of water hyacinth (DWH) was successfully achieved through a combined acid-thermal pretreatment (sulfuric acid 3% *w/w* and 121 °C) and an enzymatic hydrolysis. Optimal conditions for thermochemical pretreatment were: 0.15 mm of particle size, 50 g DWH/L $H_2SO_4$ (3% *w/w*) of solid load and 20 min of reaction. By using these optimal conditions, 277.04 ± 1.40 mg RS/g DWH were obtained. In the study of enzymatic hydrolysis of pretreated biomass of water hyacinth (PBWH), six commercial enzymatic complexes were tested. Among them, NS22118 (beta-glucosidase) and Cellic® CTec2 (cellulase and hemicellulose complex) achieved higher saccharifications. By using a NS22118 1:10 dilution and a mix of NS22118 1:10 dilution and Cellic® CTec2 1:50 dilution (in a 1:1 ratio), 914.91 ± 51.06 and 1175.67 ± 84.57 mg RS/g PBWH were released, respectively. Chromatographic analysis revealed that as a result of thermo-chemical pretreatment and enzymatic hydrolysis of water hyacinth biomass, mostly pentoses and hexoses were liberated, respectively. Therefore, it follows that hemicellulose is basically hydrolyzed in thermo-chemical pretreatment and cellulose in the enzymatic hydrolysis. Work is ongoing to produce carbohydrolases from recently isolated fungal strains, through solid state fermentation processes, in order to have more specific enzymes to hydrolyze water hyacinth biomass and to reduce costs in the saccharification processes. These experiments will be published in the near future.

**Author Contributions:** Conceptualization, J.C.; methodology, A.E.O.-G.; software, A.E.O.-G.; validation, J.C. and Y.G.-G.; formal analysis, A.E.O.-G.; investigation, A.E.O.-G.; resources, J.C. and Y.G.-G.; data curation, E.R.-B.; writing—original draft preparation, E.R.-B.; writing—review and editing, J.C. and E.R.-B.; visualization, E.R.-B.; supervision, J.C.; project administration, J.C.; funding acquisition, J.C. All authors have read and agreed to the published version of the manuscript.

**Funding:** This research was funded by the State Council of Science and Technology of Jalisco (COECYTJAL), through the research project grant FODECIJAL 8155-2019 "Sustainable use of water hyacinth that thrives as weeds on the Santiago River".

**Institutional Review Board Statement:** Not applicable.

**Informed Consent Statement:** Not applicable.

**Conflicts of Interest:** The authors declare no conflict of interest.

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
