# Peer review of "Thermochemical and Enzymatic Saccharification of Water Hyacinth Biomass into Fermentable Sugars"

_processes, doi:10.3390/pr10020210_

Round 1

Reviewer 1 Report

The manuscript is interesting, was well planned and realized. I do not have further comments, and i recommand the publication of this research manuscript.

Author Response

We appreciate the participation of Reviewer 1

Thanks

Reviewer 2 Report

Comments:

General

  1. Please write full names of commercial enzymes e.g. Cellic® CTec2 (Novozymes), endo -xylanases NS22083 (Novozymes), especially when mentioning for the first time in the text (abstract)

Specific

Abstract

  1. Lines 12-15: „Water hyacinth (WH) is an aggressive, free-floating perennial aquatic plant that is considered a pest due to its rapid grown rate “high biomass yield,” and detrimental effects on environment, human health and economic development WH is not a food source for humans or animals.“

Rewrite the sentence; it is hard to follow the text.

  1. Line 20: „… from WH biomass were stablished.“

Did you mean „biomass were established?“.

  1. Write full names of commercial enzymes.
  2. Lines 21-22: Please write the amount of RS on specific type of lignocellulosic biomass e.i. pretreated WT or raw WT.
  3. Line 26: Please delete „ respectively“; the word is unnecessary at the end of the sentence.

Introduction

  1. Lines 30-65

In the first part of the Introduction section, the authors have extensively written about the measures applied to put down under control the growth of WT.  I would suggest shortening this part since the focus of the manuscript is not control of WT growth but the pretreatment of this biomass and enzymatic hydrolysis (lines 30-59 versus 59-82). More text on pretreatment methods used for this type of biomass, enzymatic hydrolysis and cultivation of microorganisms on released sugars is missing.

  1. Line 59: „Water hyacinth is composed on dry basis by cellulose…“

The sentence is hard to follow; please consider rewriting it.

Materials and Methods:

  1. Please chech the names of the protocols used in biomass analysis and correct them in the text.
  2. Please rewrite the following sentence; they are unclear

Lines 113-114. „One hundred milliliters of sodium chlorite at 1.5 % and glacial acetic acid were added to 2 g of sample.“

Lines 120-121: „Reducing sugars (RS) were assayed using the dinitrosalicylic acid (DNS) reagent method [39].“

Lines 133-134: „Dry water hyacinth (WH) was acid-thermally pretreated using 3% w/v H2SO4 at 121°C“

  1. Line 142 „Data represent mean and standard deviation (n = 6).“
  2. Table 1. Instead of „B-glucosidase“ write „beta-glucosidase“
  3. Table 1. Please correct the activities units for different hydrolytic enzymes

Results and discussion:

  1. Lines 196-198:

„Water hyacinth (WH) plants were twice harvested in Lake Chapala (Jalisco, Mexico), in February and August 2020. Plants were washed and sectioned in leaves, stems and roots. Plant fractions were dried (at 80°C) and dry matter percentages are reported…“

It is not necessary to repeat facts that were already mentioned in the Materials and Methods. Therefore, please consider deleting the text.

  1. Table 3. The total content of different fractions in WT leaves exceeds 112 % (g/g). Please explain what might be the reason for overestimating the content of specific fractions? (please include the text in the section Results and discussion).
  2. Table 3. Please use the singular for the word leaves instead of plural (other WT fractions were written in singular)
  3. Table 3. The cellulose and hemicellulose content was not determined in root fractions of WT. Since the content of other fractions (ash, HE, HWS and ASL) was lower than in leaf and stem, it can be expected that cellulose and/or hemicellulose would be present at higher levels. Why were the authors omitted this analysis?
  4. Line 233: „… fewer furfural compounds…“ Furfural has no plural. Furfural is a chemical compound with a specific composition C4H3OCHO. Please correct the text.
  5. Lines 240-242: „It is important to note that in the lignocellulosic structure of the water hyacinth biomass, hemicellulose is preferentially hydrolyzed by the chemical pretreatment.“

It is well known that hemicellulose hydrolyses faster than cellulose in acids environment due to its chemical characteristics (See:  Xu, W., Grénman, H., Liu, J., Kronlund, D., Li, B., Backman, P., ... & Xu, C. (2017). Mild oxalic-acid-catalyzed hydrolysis as a novel approach to prepare cellulose nanocrystals. Chem-NanoMat 3: 109–119.). Please correct accordingly.

  1. Lines 251-253: „This carbohydrate fraction disappeared at the highest time reaction assayed (60 min at 121°C) and it was converted equivalently into more monosaccharides“

Decreasing the sugar concentration was probably degradation at high temperature by strong acid. Please consider rewriting the sentence.

  1. Lines 257-260: „A yield of 277.04 ± 1.40 mg RS/g DWH was calculated for optimal conditions of the chemical pretreatment (0.15 mm of particle size, 50 g DWH/ L of H2SO4 3% of solid loading 259 and 20 min of reaction time).“

When writing the RS yield per g of the dry weight of biomass, it should be mentioned that yield was calculated on pretreated biomass and not raw WT.  The yield of RS on untreated/raw biomas is significantly lower.

  1. Lines 272-274. „Initially, a dilution 1:100 for each commercial enzyme was assayed for two solid loadings (50 and 125 g PBWH/ L of diluted commercial enzymes). Reaction conditions were 50°C, pH 4.8 and 140 rpm for 24 h.“.

The method for the enzymatic reaction was already described in the Materials and Methods. Therefore is not necesarily to describe it again. Please consider replacing the text.

Furthermore, cellulolytic enzyme activity and load rates are usually presented in activity units per gram of substrat (glucan, xylan, etc.). Please rewrite the enzyme loading in standard units (e.g. FPU/ g glucan) instead of dilution rate (e.g. 1:1000).

  1. Table 4. Please correct the names of commercial enzymes.
  2. Table 5. Please consider writing the enzyme loading in FPU/g of glucan.
  3. Table 4 and 5. The concentrations of reducing sugars and sugar yields are unrealistically high. Enzymatic hydrolysis of 50 g of pretreated WT/L gave almost 50 g/L of reducing sugars and 1175 mg of RS per gram of pretreated lignocellulosic biomass. Pretreated biomass contains approximately 50 % of carbohydrates, meaning that the yield of sugars is almost 200 %. Please check the results in those Tables. Please correct the discussion of these results.

Conclusions:

  1. The conclusion should be rewritten accordingly to the corrected results.

Reviewer 3 Report

Authors submitted a manuscript on interesting option of valorization of water hyacinth (WH) biomass. Following issues are to be addressed due during the revision of the paper:

  1. There are some language errors just in the beginning of Abstract - L13-14. Please add ",", "." or start a new sentence, when required. In line 232 it would be better to write "smaller number of furfural compounde (...)" rather than "fewer"; also in line 274 "Table shows".
  2. In the introduction section, L66-76, Authors should at least list other pre-treatment methods than acid/acid-thermal pre-treatment. This can be done on the basis of references such as: DOI: 10.1016/j.biortech.2021.125235; DOI: https://doi.org/10.3390/molecules23112937; DOI: 10.1016/j.jclepro.2021.129038. Additionally, the selection of the acid pre-treatment should be more justified, in relation to other methods.
  3. In the introduction section, Authors should clearly state the novelty of this paper in relation to other available papers on the processing of WH.
  4. Materials and Methods section must be thoroughly revised and supplemented with the descrption of employed apparatus/machines. Please provide information on hot air oven (model, producer) (L92); milling machne (L93-94); spectrophotometer (L124). 
  5. Section 3.1 is a typical characterisation of main biomass fractions, while in the topic there is an inforation on heavy metals. The results are missing and would be of great importance, e.g. from the perspective of phytoremediation potential during one growing season, but also the influence of metals on the saccharification yield. Please supplement these data according to the title of the paragraph.
  6. Editing of a table captions is needed for Table 2 (L208-209).
  7. Why temperature was not evaluated as an important parameter of thermochemical pre-treatment? This is a crucial parameter, also from the viewpoint of energy/economics.
  8. Why the Authors did not use e.g. surface response methodology and other related experimental design approach to investigate the influence of selected process parameters on the saccharification yield? 
  9. Authors are encauraged to include some, even simple and basic, economic/energetic evaluation of the proposed pre-treatment conditions i.e. whether the conditions are economically favorable or other conditions would be better, ensuring comparable saccharification efficiency.
  10. Future research perspectives on valoriation or pre-treatment of WH should be addressed to in the conclusions section.

Reviewer 4 Report

Abstract:

Line 12ff:

Water hyacinth (WH) is an aggressive, free-floating perennial aquatic plant that is considered a pest due to its rapid grown rate “high biomass yield,” and detrimental effects on environment, human health and economic development WH is not a food source for humans or animals.

Comment: awful writing

Line 59:

Water hyacinth (specify the DM in %) is composed on dry basis by cellulose (18.2 to 19%), hemicellulose (48.7 to 50%), lignin (3.5 to 3.8%) and crude protein (13 to 13.5 %) [21].

Comment: What is the dry matter (DM) content? (it is mentioned in Table 2. However, you should specify the mean)

Line 140 and line 164:

Suspensions were centrifuged at 4000 rpm for 15 min.

Comment: What was the centrifugation number?

Table 1: check the abbreviations

Line 88ff and Line 196ff:

  1. Line 88ff

Water hyacinth plants were collected in February and August 2020 from the shores of Lake Chapala in the Mexican State of Jalisco (20°17'24.5"N 103°11'44.3"W). Plants were washed with tap water to eliminate dirt and then fractioned in three parts: root, leaves and stems. The wet plants were weighed, registered for each fraction, cut into smaller pieces, and dried at 70°C in a hot air oven for 72 h until constant weight was reached.

  1. Line 196ff:

Water hyacinth (WH) plants were twice harvested in Lake Chapala (Jalisco, Mexico), in February and August 2020. Plants were washed and sectioned in leaves, stems and roots. Plant fractions were dried (at 80°C) and dry matter percentages are reported in Table 2.

Comments:

Do not repeat.

Do not contradict (70°C or 80 °C?)

Table 2:

Comment: Would you explain how you did calculate the “Dry weight (%)”. The total of “Dry weight(%)” must also give 100. Something is wrong with your “Dry weight(%)” calculations.

Line 249:

In addition, at time reactions of 10, 20 and 40 min (at 121°C), a fraction of other carbohydrates different from monosaccharides were revealed by chromatographic analysis (certainly oligosaccharides).

Comment: What is meant by “time reactions”?

Line 158ff and Line 266ff:

Line 158ff

Pretreated biomass of WH (PBWH) was saccharified, using six commercial carbohydrolases complexes (from Novozymes): NS22086, NS22083, NS22118, NS22119, NS22002and CTec2 (Table 1). Initially, enzyme complexes were diluted 1:100 in citrate buffer (0.1M pH 4.8), containing sodium azide (20 mg/L).

Line 266 ff:

Six commercial preparations of carbohydrolases from Novozymes, comprised of cellulase (NS22083), endoxylanase (NS22086), β-glucosidase (NS22118), arabinase, β-glucanase, cellulase, hemicellulase, pectinase and xylanase complex (NS22119), β-glucanase and xylanase complex (NS22002) and cellulase and hemicellulase complex (CTec2) were evaluated for their hydrolytic capacity of the pretreated biomass of water hyacinth (PBWH).
Initially, a dilution 1:100 for each commercial enzyme was assayed for two solid loadings (50 and 125 g PBWH/ L of diluted commercial enzymes).

Comments:

  • Do not repeat.

Line 287:

….. representing the 13.1%, 15.8% and 15.9%, respectively,

Comment:

What is meant by this specification?

Round 2

Reviewer 2 Report

The manuscript has been revised according to the suggestions and comments.

Author Response

Again, the authors appreciate your contribution to improve the writing of this Paper.

Reviewer 3 Report

The manuscript was revised in accordance to the reviewer's comments, but without addressing one of them. In my opinion, the manuscript may be accepted for publication providing that Authors supplement the intruction section with a clear statement of the novelty of this study.

Author Response

Dear Revisor 3,

We have considered that it is necessary to emphasize the originality of our Paper.

Our Paper offers two novelties regarding the saccharification of Water Hyacinth biomass:

1) Actually, the reported hydrolysis conditions (solid loading, hydrolysis temperature, heating time and particle size) of the thermochemical pretreatment of WH biomass; and consequently, the obtained reducing sugars yields are variable. Those reported ranges of hydrolysis conditions were taken into consideration, being optimized in this work.

2) Studies in enzymatic hydrolysis of pretreated WH biomass has not yielded complete conversion to monosaccharides. Our study deals precisely with the complete hydrolysis of the WH biomass (pretreated).

Both contributions of the present study were highlighted in the Introduction section. You can find that information on lines 82-86 and 92-93.

Again, the authors appreciate your contribution to improve the writing of this Paper.